# The Role of Major Transcription Factors in Solanaceous Food Crops under Different Stress Conditions: Current and Future Perspectives

**DOI:** 10.3390/plants9010056

**Published:** 2020-01-02

**Authors:** Lemessa Negasa Tolosa, Zhengbin Zhang

**Affiliations:** 1Key Laboratory of Agricultural Water Resources, Hebie Laboratory of Agricultural Water Saving, Center for Agricultural Resources Research, Institute of Genetics and Developmental Biology, Shijiazhuang 050021, China; lami.negasa@gmail.com; 2University of Chinese Academy Sciences, Beijing 100049, China; 3Innovation Academy for Seed Design, Chinese Academy of Sciences CAS, Beijing 100101, China

**Keywords:** DOF, WRKY, NAC, HSF, MYB, ARF, ERF transcription factors, solanaceous, stress, genes

## Abstract

Plant growth, development, and productivity are adversely affected by environmental stresses such as drought (osmotic stress), soil salinity, cold, oxidative stress, irradiation, and diverse diseases. These impacts are of increasing concern in light of climate change. Noticeably, plants have developed their adaptive mechanism to respond to environmental stresses by transcriptional activation of stress-responsive genes. Among the known transcription factors, DoF, WRKY, MYB, NAC, bZIP, ERF, ARF and HSF are those widely associated with abiotic and biotic stress response in plants. Genome-wide identification and characterization analyses of these transcription factors have been almost completed in major solanaceous food crops, emphasizing these transcription factor families which have much potential for the improvement of yield, stress tolerance, reducing marginal land and increase the water use efficiency of solanaceous crops in arid and semi-arid areas where plant demand more water. Most importantly, transcription factors are proteins that play a key role in improving crop yield under water-deficient areas and a place where the severity of pathogen is very high to withstand the ongoing climate change. Therefore, this review highlights the role of major transcription factors in solanaceous crops, current and future perspectives in improving the crop traits towards abiotic and biotic stress tolerance and beyond. We have tried to accentuate the importance of using genome editing molecular technologies like CRISPR/Cas9, Virus-induced gene silencing and some other methods to improve the plant potential in giving yield under unfavorable environmental conditions.

## 1. Introduction

In their natural environments, plants are persistently and simultaneously confronted to diverse biotic and abiotic stresses, whose impacts are enhanced by climate change resulting from anthropogenic activities because of dramatic population increases which resulted in restraining the water availability and upsurge the temperature [1]. The incidence of global drought is expected to grow beyond 20% by the end of this century [2]. The potential outcomes of these changes are low rainfall (water deficiency), increased marginal land and low water use efficiency, all of which would impact Solanaceous crops. Extreme conditions such as water shortage (drought), salinity, extreme temperatures (low or high) and diseases are the most impactful environmental stresses in terms of plant growth rate, crop production quantity and quality utmost [3]. At the cellular level, these environmental extremes may lead to cell injury or damage by generating reactive oxygen species (ROS) and change in temperature that directly or indirectly increases the viscosity of cellular contents, cell shrinkage (wilting, bleaching) and chlorophyll degradation that finally ensures the demise of the plant growth [4]. 

The major Solanaceous food crops such as tomato (*Solanum lycopersicum*), potato (*Solanum tuberosum*), eggplant (*Solanum melongena*) and hot pepper (*Capsicum annuum*) are in high demand worldwide for their nutritional and medicinal values, high vitamin content and psychotropic effects i.e., solanaceous crop can be used as a source of drugs and energy [5]. However, Solanaceous crops are highly vulnerable to drought and salt stress due to their body mass (fruits and tubers), succulence and high-water requirement particularly during the reproductive stages i.e., flowering, fruiting and seed development [6,7,8,9]. Not only abiotic factors but also biotic stresses are severely impacting the yield and quality of these crops. The most common pathogens that attack the solanaceous crops are Tobacco mosaic virus (TMV), Pepper mottle virus (PepMov), Tomato spotted wilt virus (TSWV), *Phytophthora capsici*, *Fusarium* species and *Collectricum* species [10]. These above-mentioned stresses led to a change in protein-protein interactions, protein aggregation, and denaturation, which collectively disrupt the plant life cycle and interfere with their survival and physiological activities [11]. 

To cope with different biotic and abiotic stresses, plants have developed different defense mechanisms during their life cycle (growth, development and reproduction). Transcription factors (TFs), which is also known as trans-acting factors, are proteins that identify and bind to specific DNA sequences (often in the promoter region) and control cellular processes by sustaining and regulating the expression of target genes during the transcription processes [12]. This means that, regulation of gene expression at the transcriptional level is mainly mediated by the specific recognition of cis-acting promoter elements by specific DNA-binding transcription factors which has capable of turning on/off the gene when the plant gets exposed to the stress. It has been mentioned in different research finding that transcription factors have principal roles in plant immunity and many other biological functions by controlling the target gene function [13]. Most of the time, structurally, transcription factor has two different functional domains which take place in DNA binding and transcriptional activation and/or repression particularly when plant faced stresses [14]. These two domains together with other motifs participate in the activation and/or repression of the transcription process in response to endogenous and exogenous stimuli, which in turn mediate different physiological and biochemical processes [14]. 

Moreover, up-to-date more than 60 transcription factors have been identified in plants using bioinformatics tools, next generation sequencing and other methods in plant genome-wide association and other molecular studies [15]. But in this review, we have focused on the major and most studied transcription factors in solanaceous including: WRKY, DOF, MYB, bZIP, ARF, ERF, HSF and NAC transcription factors. These transcription factors are very important and play role in various biochemical and developmental processes to activate or repress different plant metabolic activities depending on the demands of the plant basically during the stress conditions [16]. 

## 2. Molecular and Physiological Mechanisms of Stress Tolerance in Plants

Research in tomato and potato have underscored that plants use different methods to hold out the environmental stresses such as drought, salinity, oxidative stress, cold, heat, and diverse diseases that affect the plant growth and development in different ways. Given the sessile nature of plants, environmental stresses may lead to sub-optimal growth conditions, imposing plasticity in different metabolic pathways that allows plants to function while resisting, tolerating or recovering from the stress conditions. As shown in Figure 1, the plant response to different stresses (primary and secondary) is a complex and different processes which involves the activation of various pathways, gene interplay and different molecular ‘crosstalk’ [17]. On the other hand, when the plant gets exposed to multiple stresses, they have got the ability to protect themselves from one or more stresses which directly help in identifying the most resistant and tolerant varieties in developing a plant of desired characteristics. In most cases, biotic stress resistance genes mediate pathogen recognition through stimulation of a signal transduction pathway that leads to the manifestation of resistance level. The molecular understanding of biotic stress response has relied heavily on the manipulation of single genes, such as over-expression or mutation analysis, to characterize the diverse components, including the sensor, the signal transduction factors and genes that produce antimicrobial factors. 

The complexity of the abiotic stress response arises from its polygenic nature and the coupling of signal transduction events with the production of stress response factors when the plant gets exposed to stresses. Noticeably, abiotic stress factors such as drought(water stress), water logging( excessive water), salinity, mineral toxicity, variation in temperature(frost, cold, heat) are unfavorably impact the physiological components of plants by altering their metabolism, growth and development status as plants are sessile (immobile) in nature and cannot escape from the environmental cues [18]. As it can be seen from Figure 1, the first phase in abiotic stress response in plant cell is recognizing the stress stimulus through sensors or receptors confined mostly at the cell membrane followed by signal transduction events involving second messengers, e.g., cytosolic calcium leading to TF activation. TFs then bind to activate stress-response genes, resulting in the synthesis of stress-related gene components and the expression of stress tolerance genes. To some degree, these complex cellular responses can be classified into three different phases during the abiotic stress response: alarm, resistance and exhaustion [19].

Environmental stresses (primary and secondary) which have been mentioned in the Figure 1 causes the activation of different signaling pathways and transcription control processes. Biotic stresses which are caused by pathogens were included under primary stresses. Stress-response mechanisms are initiated to re-establish plant cellular homeostasis along with the defense against stress and mending of damaged proteins and membranes. It shows that transcription factors and osmoprotectants aid in withstanding the stresses during plant growth and development. The re-establishment of cellular homeostasis is critical for the ability of plants to gain tolerance or resistance to stress.

## 3. Reverse Genetics: Improving Plant Traits against Stresses Using TFs

In plant breeding and improvement, creating variation in the gene pool is an important way of improving plant traits and ensuring genetic diversity at prominent level. The conventional method of crop improvement has many bottlenecks such as mutations, gene duplications/deletions and chromosomal rearrangements and etc. However, nowadays, there is a conspicuous promise for addressing the evolving challenges in plant production because of the ongoing climate change. To challenge the climate change which is the big issue for the world right now, the currently emerged molecular technology, CRISPR/Cas9 may play pivotal role in improving the traits of plant against the climate change. It is sequence-specific genome editing technology which is becoming a powerful tool in improving agronomic trait of crops and play role in ward off pathogens and minimizing the risk of different stresses even at multiplex genome editing techniques either by adding or removing the gene of interest [20]. Most importantly, it improves plant’s valuable traits as climate change is affecting the plant growth and development parameters such as seed germination, photosynthesis, source-sink relationships during growth, cell division, enzyme activities, and secondary metabolites production (it helps in abiotic stress tolerance) [21]. In recent years, many resistant and tolerant genes against stresses have been isolated from major food crops including solanaceous and there are many important genes involved in stress tolerance but their function is not well known yet. Therefore, studying the function of genes using this technology is very much important because developing transgenic plant is time consuming, quite expensive and laborious compared to reverse genetics which is available these days.

Basically, the CRISPR/Cas9 can cut or add the target regions of genes directly in a DNA sequence in very specific way. This means that, it is easy to generate stable and heritable mutations without affecting the existing valuable traits in plants or other organism. In this way, it is possible to develop the homozygous modified transgene-free plants in only one generation and its stable transmission to successive generations which can save time immensely [22,23]. Taken together, CRISPR-Cas9 system has three very relevant stages in response to the invading foreign DNA: (i) acquisition/adaptation is stage at which the invading DNA is known and a spacer sequence derived from the target DNA is inserted into the host CRISPR array; (ii) expression stage is a procedure when expression of Cas9 protein and transcription of CRISPR array into a precursor RNA transcript (pre-crRNA) is undertaken. A non-coding CRISPR RNA (crRNA) then hybridizes to the pre-crRNA and Cas9 protein to produce mature RNA (crRNAs); and (iii) interference stage then take place when the mature crRNA guides the Cas9 protein to recognize the DNA target, leading to the cleavage and degradation of the invading foreign DNA [24,25]. Most recently, the review paper written on CRISPR-Cas9 has shown the developmental roles and different environmental stresses tolerance and resistance genes of tomato such as *MLO1*, *DMR6*, *MAPK3*, *BZR1*, *CBF1*, *AGL6*, *IAA9*, *MPK20*, *GAD (2*, *3)*, *BOP (1*, *2*, *3)* [26]. As a result, knocking out genes that confer undesirable traits is the main role of CRISPR/Cas9. By the same token, traits such as quality of the crop, yield, biotic and abiotic stress resistance, developing hybrid seed can be achieved using CRISPR/Cas9 system too. Regarding this, the genes mentioned in this manuscript can be used in stress studies using CRISPR Cas9 system in further studies to develop more resistant and tolerant varieties.

For instance, in tomato, chilling stress is the main restraint because it is tropical in origin and very-sensitive to chilling stress, which can restrict the flowering and fruiting stage. But, currently, it has been revealed that the mutant that has been made using the CRISPR/Cas9 gene system to understand the functional relationship between *SlCBF1* protein and chilling stress tolerance have shown that it was positive, meaning that it helps in tolerating the chilling stress. On the other hand, the highly conserved CBFs are cold-response system components found in many plant species. Therefore, it was concluded that the generated *SlCBF1* mutants using the CRISPR/Cas9 system has shown that *SlCBF1* protein profoundly helped tomato in chilling stress tolerance [27]. Alternatively, *SlMAPK3* gene is responsible for drought in tomato, the knockout mutants produced by CRISPR/Cas9 has shown the same thing on the drought stress [20,27]. According to the report of [28], *MLO1* gene is also responsible for powdery mildew vulnerability just that the mutant generated by CRISPR Cas9 has shown tolerance to the disease comparing with the untreated tomato plant.

The other genome editing technology is Virus-induced gene silencing (VIGS) which is also the most valuable tool to understand the functional confirmation of stress-responsive genes which shows high transcript profile (high expression level) when the plant gets exposed to different kind of stresses. Most importantly, the TRV-VIGS-mediated silencing method is very imperative in solanaceous crops for studying the gene function at the transcription level. For instance, it has been pinpointed that a GLUTAREDOXIN gene, *SlGRX1* is responsible for drought stress in tomato as the satellite-virus-based vector, DNAmβ was used for confirmation at physiological level by measuring chlorophyll and relative water content (RWC) which both of them have shown low content compared to the control plant in addition to phenotypic differences [29]. On the other hand, an extracellular PEROXIDASE 2 (*CaPO2*) gene in *Capsicum annum* is responsible for osmotic stress in which bleaching was displayed more than the control plant. The research that has been done on the silencing of the *ABI3/VP1* transcription factor (*CaRAV1*) alone or together with OXIDOREDUCTASE (*CaOXR1*) was shown the same thing in the plant [30,31]. Most strikingly, we want to point out that it is better if each transcription factor is evaluated and studied under different stress conditions to boost and improve the yield which helps in utilizing the marginal land where water deficiency is a problem. 

## 4. DOF in Solanaceous Food Crops

The plant-specific multigene DOF (DNA-binding one finger) transcription factor family is highly involved in many growth and developmental processes such as phytohormone regulation, light signaling, and seed germination. It plays major role in assisting protein-protein interaction in addition to aiding in DNA-binding activities [32]. Importantly, DOF TFs also interact with non-histone nuclear high-mobility group (HMG) proteins [33,34], which accelerate and facilitate various DNA-related biochemical activities including replication, transcription, recombination and DNA repair when an error has been made. 

DOF proteins can physically cooperate with each other or modulate transcription in the genome at defined sites [34] and can also interact with other classes of TFs, such as basic leucine zipper (bZIP) TFs [35], zinc finger protein (ZFP) TFs [35], WRKY TFs [36,37] and MYB TFs [38,39]. The characterization of stress-related genes and their regulation at the transcriptome level is critical for target gene selection and the engineering of stress-tolerant or resistant transgenic plants. Some of the Dof DNA-binding proteins can interact with a specific sequence in the very commonly used cauliflower mosaic 35S promoter, and the sequence was understood to function as a cis-regulatory region in directing transcription [40]. The structure of Dof transcription factor has about 200–400 amino acids in length with two major domains: (1) an N-terminal Dof domain with a highly conserved 50–52 amino-acid domain, and (2) a variable domain at C-terminal end for transcriptional regulation (gene expression activities) of different metabolic activities [34,41]. 

The Zinc finger motif has 29 amino acid residues and another 21 amino acid residues at its C-terminal region just that the residues can participate between Dof domain and DNA interactions. Because of the strong similarity among the Dof DNA binding domain, it is characterized by a C2C2-type zinc finger motif that distinguishes the specific regulatory elements of four nucleotide bases of “AAAG” or “CTTT” in the promoters of the specific genes as it has been studied in vivo and in vitro experiments [42,43,44]. One exception from this principle is the pumpkin Dof protein named as AOBP, which distinguishes an AGTA repeat sequence [45]. Basically, in the same way to other zinc fingers, the DNA-binding domain has dual functions: DNA-binding and protein-protein interactions [32]. It also plays a very imperious role in the regulation of secondary metabolic activities like biosynthesis of glucosinolates and flavonoids, which contribute to the defense mechanism when plant faced their pathogen [46,47]. The identifying feature of Dof protein families that distinguishes it from other TFs is that it has four cysteine residues in the conserved Dof domain with a single C2–C2 type zinc finger, which binds to (A/T) AAAG or its inverse CTTT (A/T) sequence as the recognition core [44].

TFs belonging to the same family may exhibit various activities owing to the occurrence of a transcriptional regulatory domain that may act as a repressor or activator during the gene expression [42]. The repression of gene expression typically involves the prohibition of activators from target promoters by the role of competitive binding between TFs for the alike cis-acting element [43,48]. Alternatively, gene expression repression may result from the masking of regulatory domains by the dimerization of TFs or the interaction of repression domains with TFs. Research on a Dof protein in barley (BPBF) showed that it normally activated the transcription of a putative target, but a mutation disrupting one of the BPBF cysteine residues resulted in its inactivation [49]. Nowadays, genome editing methods are helping widely to modify abiotic stress response in plants and increasing the yield under a stressed environment which can be a great contribution in minimizing the marginal land [50].

Heat maps showing the global transcription patterns of hot pepper Dof genes were generated based on publicly available RNA-seq data for the plant tissues in different developmental stages and varieties [51] (Table 1). On the other hand, the expression analysis Dof genes undertaken in eggplant varieties, some genes were not expressed (*SmeDof25* and *SmeDof4*), and others were extremely low (*SmeDof19*, *SmeDof20*, and *SmeDof22*) and high (*SmeDof17*, *SmeDof23*) which may be promising for the development of transgenic lines [52]. Similarly, the StDof gene expression profiles in potato were analyzed in leaves, shoots, roots, stolons (hooked apex stolon), swelling stolons and mature tuber. Most strikingly, the *StDof* genes were expressed in all potato tissues, although the expression levels of individual genes varied in tomato, eggplant, and hot pepper. Seventeen of 33 genes have shown relatively high expression levels in the root. In the same way, 22 *SlDofs* in tomato and 22 *GmDofs* in soybean were expressed at relatively high levels in the root [53]. Therefore, it is very important using the expressed genes in developing the transgenic plant under different stress condition. There were other DOF genes information in major solanaceous food crops [54,55,56,57,58] (Table 1).

## 5. WRKY Transcription Factor in Solanaceous

This transcription factor is widely studied in different plants including the solanaceous crops and it is considered as one of the largest and important transcription factors in connection with the environmental stresses both biotic and abiotic stresses. The WRKY protein has one WRKY domain which includes 60 amino acid residues like most the other transcription factors. Structurally, its N terminus has the WRKYGQK motif and the C-terminal has metal-chelating zinc finger motif, either C2H2 or C2HC as carbon and hydrogen play an important role in defining its structure. Therefore, the WRKY family has three groups depending on the number of WRKY domains and number of zinc finger it has in its structure. Among the three groups, Group I and II WRKY domains placed in the C- and N-terminus, respectively [59,60]. Alternatively, it is possible to say that group I and II have the C2H2-type zinc finger, and only Group III has the C2HC-type structurally. The most studies have shown that WRKY proteins have the ability which makes it specifically bind to the W-box motif (TTGAC/T) that regularly found in the promoter region of the stress related genes when the plant gets exposed to the stress conditions [14,61]. For instance, *SlWRKY30*, *SlWRKY83* and *SlWRKY75* genes in tomato have been upregulated in drought [62]. Similarly, *SlWRKY72* gene has a significant role in basal immunity and makes the plant withstand the pathogen attack when the pathogen gets conducive environment and has shown a positive role in abiotic stress tolerance. The WRKY genes studied so far in major solanaceous crop have mostly shown upregulation under different extremities like drought, salt, heat, cold and other different stresses which obstruct the plant from finishing their life cycle and producing the genetically expected yield as shown in Table 2 [63,64,65,66,67,68,69,70,71,72,73]. Therefore, these genes can be used in developing drought tolerant transgenic plant in areas where water shortage is very troublesome.

Most importantly, signaling molecules which are endogenous to plants such as SA, JA, ET and ABA play a tremendous role in regulating the signaling pathways which subsequently has capable of changing the transcription level of stress related genes and protein post-processing which as a result help as a mediators in basal defense systems [74,75]. This was revealed from the fact that *CaWRKY27* gene from *Capsicum annum*, has shown positive role in regulating the stress resistance to *Ralstonia solanacearum* infection by modulating and regulating SA, JA and ET mediated signaling pathways in tobacco (*Nicotiana tabacum*) after the plant has been exposed to the mentioned pathogen [69,75]. Thus, salicylic acid and other signaling molecule is a major signaling molecule in response to biotic stresses in cross communication with WRKY and other transcription factors which is mostly related to stress conditions. The moment the plant gets infected by a pathogen, the level of SA increases in dramatic fashion that leads to the expression of genes encoding for the pathogen proteins and activating the disease resistance gene in the plant to make it withstand the stresses. This is the reason why SA is used regularly to study how stress affects the plant growth and development [75]. Importantly, *StWRKY72* gene in potato has been also highly up-regulated under stress treatment that sounds its importance better in developing a resistant transgenic line in solanaceous crop to make sure that the developed plant can grow under a water shortage and help in marginalized land utilization utmost [62].

By and large, WRKY protein has many functions in regulation of plant growth and development in response to different stresses such as salinity, drought, heat and cold [76,77,78]. It is also involved in plant defense against different bacterial, fungal and viral pathogens which are considered as biotic stresses [79]. The research that has been done on solanaceous and Arabidopsis has explained that a WRKY protein can be used in trichomes development [80], seed development and germination [81], embryogenesis [75] biosynthesis and hormonal signal regulations and leaf senescence [82]. Importantly, to understand the complete family of WRKY transcription factor in major solanaceous crops, its gene and protein sequences can be retrieved from databases like Solgenome (https://solgenomics.net/), NCBI (https://www.ncbi.nlm.nih.gov/) and PLANTTFDB (http://planttfdb.cbi.pku.edu.cn) which are major databases for many plant transcription factors. 

Solgenome database is very common for major solanaceous crops and provides all the information for genomes, genes and protein sequences whereas PLANTTFDB database has all 60 plant transcription factors. Most prominently, the WRKY protein and gene sequences can be downloaded from the PGSC database (http://solanaceae.plantbiology.msu.edu/pgsc_download.shtml). In Table 2, the studied WRKY transcription factor in each major solanaceous food crops were summarized briefly [78,79,80,81,82,83,84,85,86,87,88,89].

## 6. MYB Transcription Factor

It is a very common transcription factor and widely studied in solanaceous crops most recently. It plays a major role in fighting against abiotic and biotic stresses in plants. Basically, it has two distinct features, an N-terminal which has a conserved region of MYB DNA-binding domain with about 52 amino acid residues and a diverse C-terminal region which is in charge of regulating the protein activities. Depending on its conserved MYB domain, it consists of four classes: 1R, R2R3, 3R and 4R-MYB proteins. Among these four classes, R2R3 constitutes the largest TF gene that studied broadly under different environmental extremities [83].As shown in Table 3 [84,85,86,87,88], among the isolated MYB transcription factor in eggplants, *SmMYB1* and *SmMYB6* genes were positively regulated anthocyanin biosynthesis which in turn plays a very vibrant role in different growth regulation and promotion mechanisms [84]. The significant accumulation of anthocyanin in eggplant because of the overexpression of the *SmMYB1* gene revealed that this gene has a key role in anthocyanin production. On the other hand, anthocyanin structural genes were upregulated in eggplant which naturally correlated with the color in its fruits and other parts of the plant [85]. By the same token, the *StMYB113* gene which is homologous to *AtMYB113* gene plays a significant role in positively regulating the phenylpropanoid metabolism in *Arabidopsis thaliana* [86]. A recent study suggested that anthocyanin structural genes and the R2R3-MYB expression level positively regulated by light exposure which in turn help in anthocyanin biosynthesis enormously [87,88]. On the other hand, *PhMYB27* and *PhMYBx* genes of petunia are also the two most indicators of R2R3-MYB, and R3-MYB repressors, respectively [89]. 

Additionally, R2R3 MYB family aids in regulating the expression of different catalytic enzymes such as the pathway of anthocyanin, a chemical which makes the eggplant colorful and advantageous for human health [90,91,92]. However, several MYB transcription factors have capable of negatively regulating or repressing the expression of anthocyanin biosynthesis structure genes which can be checked by the reduction of color it shows after the gene repression. Most strikingly, it is possible to pinpoint that MYB regulates anthocyanin biosynthesis structure genes negatively through the inhibition of the formation of the MBW complex [93]. In eggplant, phenylpropanoids are also the major secondary metabolites in its fruits, the palatable part. Fundamentally, chlorogenic acid (CGA) which has more than 70−60% of total phenolic in its tissues is considered as the main phenylpropanoid metabolite in most of the solanaceous crops as it plays a pivotal role in cardiovascular diseases treatment [94,95,96]. 

It is very important to study CGA molecules in other vegetables and fruits to know much more about its healthy functions. Tomato and potato’s skin and flesh tissues are characterized by its distinctive metabolite content whose phenylpropanoid profile differs in different tissue parts demonstrating that their level of accumulation is tightly maintained [96]. Therefore, this molecule is highly regulated by MYB transcription factor in plants including solanaceous crops [97]. Hence, the productions of phenylalanine-derived compounds are largely regulated by R2R3-MYB proteins, which are noticeably considered as the largest class of secondary metabolism regulators [98].

For instance, in tomato, MYB transcription factor encoded by two paralogs of tomato’s anthocyanin genes; *SlANT1* and *SlAN2* which has the potential of regulating fruit’s anthocyanin pigmentation and color [99]. These two genes induce anthocyanin synthesis in its fruit when their expression is driven by the 35S promoter (35S: SlANT1 and 35S: SlAN2) and both of them have shown the high expression levels of SlDFR which has capable of encoding a key enzyme in anthocyanin biosynthetic pathway. The study that has undertaken on the silencing of the *SlAN2* gene using virus-induced gene silencing has normally shown the anthocyanin biosynthesis in the fruits and the transcription of the *SlDFR*, *SlAN1*, and *SlJAF13* levels were reduced tremendously. But the change was not observed following the silencing of the *SlANT1* gene [100]. Anthocyanin is a natural antioxidant compound that has the potential to protect leaves from high light intensity, irradiation and different stressful conditions that hinder photosynthetic efficiency at large [101]. Moreover, the two genes play a very crucial role in enhancing anthocyanin accumulation in tomato’s fruit which clearly shows that both MYB TFs activate anthocyanin biosynthetic genes. These two genes can add values to tomato, eggplant and hot pepper in developing varieties that have attention-grabbing skin colors to satisfy consumers comparatively. The two MYB important genes, *SlMYB7*-like and *SlMYB48*-like are also positively regulators of anthocyanin synthesis in tomato and targets of using miR858 is that it helps as a negative regulator of the same pathway of anthocyanin synthesis [98,102,103]. 

The expression of miRNAs has been regulated most of the time during the pathogen infection or when the plant gets attacked by the pathogen. Most importantly, miRNA58 regulate the expression of numerous genes taking part in a specific pathway including the phenylpropanoid pathway which plays major roles in the production of antifungal compounds to increase the plant’s ability to stand against pathogen infection massively [104].The most recent research that has been done on chilli pepper revealed that the *CaMYB31* gene regulates capsaicinoid and its pungency. The gene silencing system (Virus-induced gene silencing) has been also used to understand the function of *CaMYB31* gene by suppressing it. After the *CaMYB31* gene was silenced, the plant has shown a substantial decrease in capsaicinoid accumulation compared to the control chilli pepper plant [88]. The information provided about MYB genes functions in major solanaceous crop was shown in Table 3.

## 7. Heat Shock Transcription Factor

This transcription factor was discovered 30 years ago and participates in transcriptional activation of the genes controlled by thermal stress (heat stress), which has capable of encoding heat shock proteins i.e., it shows a high level of gene expression under thermal stress. On the other hand, the expression of heat-shock genes increases when the plant gets exposed to thermal stress which results in the rapid accumulation of heat-shock proteins (HSPs). Understanding heat transcription factor started when it was first isolated from tomato [105] and has shown a remarkable role in helping the plants including Solanaceous in withstanding heat stress which is literally between 5–10 °C differences above the normal growing temperature of the plant. The fact that global warming is increasing the marginal land and posing a threat to the production of very important food and cash crops including solanaceous, it is very important to study heat shock transcription factors which can surely make the plant withstand the heat stress at possible levels. 

Therefore, when the plant gets exposed to high temperature (thermal stress), heat transcription factor which is also known as the central regulator of heat shock stress response, regulate the expression of many heat-stress-inducible genes at transcription level by recognizing the conserved binding motifs (heat stress element, HSE) that exist in the promoter region so that plant can withstand thermal stress. Additionally, HSPs guard the cells against thermal and other environmental stresses which at the same time participate in protein folding that helps in protein function, cell differentiation, dimensional structure and conformation [106,107]. Noticeably, it also aids the plant to withstand the high temperature and prevent the protein denaturation and cell differentiation. On the other hand, heat shock transcription factors take part in the initiation of genes responsive to different abiotic stresses such as heat, drought stress and in different chemical stresses such as Cd^2+^ and salicylate which is highly toxic to human health [108,109].

Although the solanaceous plant could survive in a different range of agro-ecological conditions, both vegetative and reproductive growth is severely affected by heat (thermal) stress which resulted in low yield and/or poor quality fruits [110,111]. Exposure to heat stress, make the heat-labile proteins denature easily and the ROS elements increase in plant cells which finally kill the plant by drying it [112,113]. Different researches have underscored that HSFs also regulate different abiotic stresses such as salt and drought and other regulatory molecules in the complex network of stress response pathways [114,115]. Most noticeably, the heat transcription factor of tomato (*HSFA1a*) has shown a unique function in master regulator for acquired thermotolerance which cannot be replaced by any other HSFs families [116]. This role of HSTs has been checked by co-suppressing tomato plant (*CS-SlHsfA1a*), in which the expression levels of HSPs were decreased under normal conditions and this plant has shown more sensitivity to thermal or heat stress. Hence, it is helpful in developing solanaceous transgenic plant to improve heat stress resistance so that it is possible to use the marginal land where thermal stress is problematic. HSF gene sequences of each major solanaceous food crops can be obtained from the Plant Transcription Factor Database [115]. The complete list of HSF gene and proteins can be identified using bioinformatics tool like Hidden Markov Model (HMM) and the conserved domain can be downloaded from the Pfam database (http://pfam.xfam.org/search). 

The protein sequences of HSF conserved domains are aligned by BLAST in the NCBI (http://blast.ncbi.nlm.nih.gov/blast.cgi) and Spud DB for major solanaceous food crops (Potato, tomato, hot pepper and eggplant) and Genomics Resources database (http://solanaceae.plantbiology.msu.edu/) with E-value of 0.001 to screen candidate Hsfs with homologous amino acids sequences. These candidate genes can be analyzed using the domain identification function of the Pfam database (E = 1.0) to remove the Hsfs without the conserved domain sequences. This helps in the multiple protein sequence alignment using Clustal W and MEGA 4.0 to remove repetitive sequences. Depending on the genome wide identification and characterization of heat shock transcription factor in solanaceous crops, the total lists of HSTs were listed in Table 4. By and large, Hsps has dual functions in both stress and non-stress conditions, which plainly indicates the role it has on the stress response and physiological response by regulating complex regulatory processes which was checked by the gene expression pattern to evaluate its function and tolerance to stresses [117,118,119,120,121,122,123].

## 8. NAC Transcription Factor

It is noticeable that the transcription factors and cis-elements function in the promoter region of stress-related genes which can prevail whether the genes are overexpressed or suppressed to take part in improving the plant’s tolerance to any stresses. Having subfamilies such as the NAM, ATAF, and OsNAC3, this transcription factor has got its name from abbreviations of the three proteins NAM, ATAF1&2 and CUC2 which has similar DNA-binding domains [124]. Therefore, NAC transcription factor is among the largest transcription factor in plants which is responsible for both biotic and abiotic factor stress conditions. NAM proteins are responsible for shoot apical meristem development in plants [125]. Two arabidopsis genes *ATAF1* and *ATAF2* have got the ability to activate CaMV 35S promoter in yeast and overexpression of *ATAF1* and *ATAF2* have regulated drought tolerance during water shortage [126,127,128]. On the other hand, CUC2 (cup-shaped cotyledon 2) is also responsible for shoot meristem initiation and aids in the formation and stable positioning of carpel margin during growth and development [129]. Two decades ago, the NAC TFs were isolated and first described in Petunia, a solanaceous crop [127]. Afterward, many studies have been done on its function against biotic and abiotic stress tolerance and growth promotion [128]. Besides, this transcription factor aids in signal transduction, innate immune systems, basal defense and systemic acquired resistance in solanaceous crops and other plant species [128]. Hence, it is possible to say that most of the defense and stress related genes are activated by NAC genes to cause multiple immune responses as long as the plants encountered different stresses [129]. 

When the plant gets attacked by the pathogen (biotic stresses), there are at least three signaling mediators such as SA, JA) and ET that can be harmonized to generate the plant defense system [129,130]. Most strikingly, NAC of tomato (*SlNAC1*, *SlSRN1*, and *SlNAC35*), eggplant (SmNAC), potato (*StNAC4*, *5*, *18*, *48*, *81*) and hot pepper (*CaNAC1*) are take part in different signaling pathways to induce local and systemic disease resistance in which most of them are discussed in Table 5. The other most important thing is that JA and ABA are responsible for abiotic stresses such as drought, salinity, cold and thermal stress [129,131]. Generally, the NAC transcription factors play such a tremendous role in seed and embryo development [132,133], shoot apical meristem formation and fiber development [127,133] and developmental processes and cell divisions [134,135]. The studied NAC transcription factors in solanaceous food crops were shown below in Table 5 [136,137,138,139,140,141,142,143,144,145,146,147,148,149,150,151,152,153,154].

The research that has been done on Arabidopsis has shown that the expression of the *AtNAC1* gene was induced by lateral root development and it is also regulated by auxin which at the same time enhances the cell division and regulates the leaf senescence [135,155,156]. Moreover, NAC TF is also involved in the detoxification of ROS during biotic stress and different defense systems [157]. Most strikingly, the *SlNAC1* gene play an important role in chilling stress resistance in tomato (4 °C), heat stress (40 °C), and different osmotic (drought) stresses. This gene also aids the plant against the mechanical wounding and other kinds of similar stresses which directly or indirectly help plant’s tolerance against these kinds of stresses [136,157,158].

## 9. bZIP Transcription Factor in Major Solanaceous Crop

It is also one of the largest transcription factors found in all eukaryotes including plants. It has a bZIP domain that regulates gene expression in response to different abiotic and biotic stresses. As the name suggests, the basic region/leucine zipper (bZIP) domain consists of persistent α-helix involving a basic region (BR) which is very important in binding the DNA followed by a C-terminal leucine zipper (LZ) motif accountable for the dimerization [159]. In plants, as shown in Table 6, this transcription factor is involved in different biological functions like pathogen defense [160] abiotic stress and hormone signaling pathways [161]. It is also taking part in plant developmental processes like seed maturation and flower development [162,163] and senescence [164]. It is a very important transcription factor because of the fact that abiotic stresses such as drought, salinity, temperatures and thermal stress adversely affect the biochemical, physiological and morphological components of the plant which in turn causes wilting, drying and finally dying of the plant. The genome wide identification and characterization of this transcription factor has not been completed in all major solanaceous food crops such as eggplant and hot pepper. Therefore, it is very important to study the genome wide identification and characterization of bZIP genes in the mentioned vegetable crops to improve the abiotic and biotic stress tolerance in solanaceous crops as shown below in Table 6 [27,165].

## 10. Ethylene-Responsive Factors (ERF) TFs

Ethylene is a recognized gaseous plant hormone that plays crucial and regulatory roles in plant growth and development. It is also functioned as a stress-related hormone required in biotic and abiotic stress responses in different ways. Hence, the Ethylene Responsive Factor family which was first isolated from tobacco two decades ago is one of the largest transcription factors that involved in the ethylene signaling pathways and lead to the expression of various defense-related genes such as pathogenesis-related (PR) genes and abiotic stress-responsive genes [166]. ERFs also play a part in plant development processes such as seed germination and different developmental processes [167]. ERF proteins are important in plant responses to both abiotic and biotic stresses by binding to multiple cis-acting elements found in the promoter regions of ET-regulated genes, including the GCC box and DRE/CRT (dehydration responsive element/C-repeat) [166,168,169]. Thus, it is characterized by the presence of a GCC-box (GCCGCC) or a dehydration responsive element/C-repeat element (DRE/CRT, CCGAC) located in their promoter sequences and a conserved 58–59 amino acids and DNA binding domain that specifically bind to GCC cis-elements [170]. It also participates in a variety of biological processes of plants, such as metabolism, growth, development and responses to different environmental stresses [171]. 

By the same token, it was reported that the ERF transcription factor family members are responsive to drought, ABA, and saline conditions and can serve as activators or repressors of ABA signaling under salt stress [172]. Jasmonate Ethylene Response Factor 1 also has a potential role in plant abiotic stress responses such as salinity, low temperature, dehydration in tomato, transgenic tobacco [173]. For this Jasmonate Ethylene Response factor, Methyl jasmonate (MeJA) is a major derivative and also it is an important endogenous regulator that plays a critical role in inducing resistance against fungal pathogen [174]. The information about ERF genes functions in major solanaceous food crops were shown in Table 7 [175,176,177,178,179,180,181,182]. 

## 11. Auxin Response Factors (ARF) Transcription Factor

Auxin is one of the most important plant hormone that plays significant roles in plant growth and developmental processes such as cell division, expansion, and differentiation. It regulates the expression of early auxin-responsive genes under different developmental processes and stress conditions. Thus, the ARF proteins are a plant-specific transcription factor, which structurally consists of three components: a conserved amino-terminal DNA binding domain (DBD), a highly conserved carboxyl-terminal domain (CTD) and a variable middle region (MR) [183]. The ARF gene family plays a crucial role in response to indole-3-acetic acid (IAA) by regulating expression of down-stream target genes. The promoter of this transcription factors have a number of conserved motif (TGTCTC) or some variant of auxin-responsive element (AuxREs) [184]. ARF genes are expressed in dynamic and different patterns during growth and development of plants and it has been proved that individual ARFs has capable of controlling distinct developmental processes [185]. On the other hand, ARFs regulate the expression levels of auxin response genes by binding to the promoters of auxin response elements (AuxREs): TGTCTC/TGTCCC/TGTCAC) [186]. Most strikingly, it can activate or repress the expression of auxin response genes by binding particularly to auxin-responsive elements (AuxREs) in the promoter regions as shown in Table 8 [186,187,188,189,190,191].

## 12. List of Major Transcription Factors in Major Solanaceous Food Crops

The genome wide identification and characterization of a major transcription factors in solanaceous food crops have been almost completed and the lists have been shown in the following Table 9. Specifically, in eggplant, transcription factors such as MYB, HST and NAC have not been completed genome wide, which should be studied to increase the chance of abiotic and biotic stress tolerance not only for eggplant but also other solanaceous crops by developing transgenic plant using the resistant genes. MYB which is a very important transcription factor also has not been completed in potato genome wide. This is very inquisitive because studying the genome-wide gene expression patterns could help the plants adapt to different kinds of environmental stresses by developing the transgenic plant or using different gene silencing technologies. The number of each transcription factor in each solanaceous crop (eggplant, tomato, potato and hot pepper) after genome wide analyses have been shown below in Table 9. The genome wide analyses of each transcription factor have been completed and “–“indicates the genome wide analysis was not done yet. Potato’s WRKY [63] Dof [58], NAC [145]; ARF [192]; ERF [181] Pepper WRKY [193] ERF [194]; Dof [51] MYB [103]; ARF [190]; NAC [195] Tomato WRKY [62] Dof [53] MYB [88] HSF [120] ARF [196] ERF [171]; Eggplant’s WRKY [197] Dof [52] MYB [84].

## 13. Measuring Chlorophyll and Relative Water Content during Gene Silencing

The effects of abiotic stresses such as drought and salinity on photosynthesis are multifaceted and are directly interrelated with stomatal closure and mesophyll limitations for the diffusion of gases, which ultimately changes the net photosynthesis process. It is therefore very significant to measure the chlorophyll content of plants using spectrophotometric analysis to know how much the plant is affected physiologically. Acetone and DMSO are the two most commonly used in the extraction of Chlorophyll a, chlorophyll b and carotenoid in plants and are measured at 665, 643 and 470 nm respectively [198]. Most of the time, the rate of excitation of chlorophyll molecules surpasses the conversion of energy in the reaction centers of PSII under stress condition, excited chlorophyll molecules then can generate singlet oxygen molecules, which resulted in promote photo-oxidation. As a result, carotenoids which are also components of PSII, take action as non-enzymatic antioxidants, they scavenge excited chlorophyll molecules and drive away energy as heat [199]. To evaluate the physiological role of a particular gene under different abiotic stress conditions (mostly drought and salinity stress), the total chlorophyll of the silenced plants and the control plants should be measured to make sure that particular gene of interest play a role in a tolerance or resistance to the defined stresses depends on the silencing efficiency [200]. On the other hand this means that the membrane damage and leaf senescence of the silenced gene plant might be higher under stressed treatments. 

On the other hand, measuring the relative water content [201] is very important in understanding how much the gene silencing is effective, most of the time using Phytoene Desaturase gene (PDS) by virus induced gene silencing or other silencing method under abiotic stress such as drought and salinity. The PDS gene which also called marker gene is used in gene silencing methods because photo-bleaching phenotype is easily detectable visually. The other important thing is that if RWC and chlorophyll is higher in transgenic line, it is an indicator of how that particular gene inserted is beneficial compared to the wild type. Thus, when the stressed plant with silenced gene has shown low relative water content and chlorophyll content (highly bleached) compared the control, it is possible to say that particular gene under study is responsible for drought of other kinds stresses. 

## 14. Conclusions and Future Directions

Genome-wide studies of transcription factors such as WRKY, MYB, DOF, HSF, bZIP and NAC in solanaceous plants play a vital role in understanding the genes responsible for different stresses i.e., transcription factors might help in boosting up the yield and developing a tolerant plant to different environmental stress conditions. A major future challenge is the mitigation of climate change effects on crop production despite reduced water availability in which the systems of drought resistance may vary depends on the climate change and soil condition with variation in temperature. The advantage of focusing on plant TFs lies in the potential production of the genetically modified cultivar in which plant stress response pathways and factors can be fine-tuned to make the plant tolerate the environmental stresses. Increasing the water use efficiency of solanaceous crops is very important particularly in arid and semi-arid areas in most developing countries like in Africa because of the fact that solanaceous crop demand more water. There are no many transgenic plants developed using TF candidate genes in solanaceous crops to the wanted extent, but it may represent a good strategy in improving the production and productivity in the area of water shortage and where the disease is very severe. 

The currently available genome editing technologies like CRISPR-Cas9, VIGS and other gene editing and silencing technologies are invaluable in improving the traits of not only solanaceous plant but also other very important food crops. This has at least two major functions: firstly, it is worth emphasizing that a single TF can regulate the expression of multiple genes to help in withstanding different stresses and secondly it may also use for developing a transgenic plant of desirable characteristics. This regulatory capacity might, therefore, be useful for improving water use efficiency and yield. The expression analyses of the mentioned eight transcription factors in different solanaceous crops by testing the upregulation in different tissues (developmental regulation) or under different abiotic or biotic treatments (stress regulation). This could promise in understating the role of each gene as the potential of each specific gene is unique in terms of environmental stresses and the conditions facing dry land agriculture. Therefore, the development of transgenic plants harboring TF genes that improve stress tolerance in field conditions represents a gap that will need to be filled by future research which would be environmentally friendly. 

There are still a lot of transcription factors that need to be characterized and the variation in each gene with its function in solanaceous crop sounds better in developing better traits. Another challenge is the difficulty of ‘pyramiding’ drought-tolerance related genes in highly heterozygous tetraploids plants like potato cultivars because most abiotic stress like drought and salinity are controlled by many genes. Therefore, TF genes need to be further studied to identify suitable candidate genes to improve different stress tolerance and water use efficiency through reverse genetics analyses, gene expression studies, transgenics and using different available molecular assays technologies to keep the food security. By and large, genome editing technologies should get better attention as climate change is ongoing and number of populations is also dramatically increasing which as a result giving attention to improve the yield in quality and quantity is so much important. 

## Figures and Tables

**Figure 1 plants-09-00056-f001:**
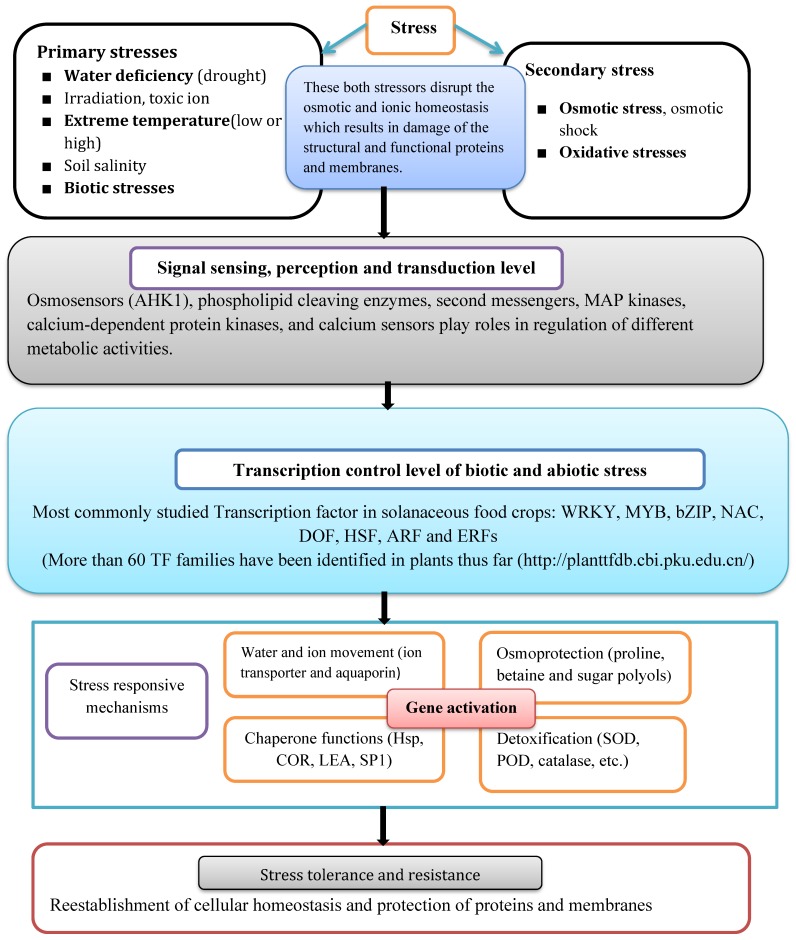
Physiological and molecular mechanism of the plant response to the stresses.

**Table 1 plants-09-00056-t001:** Dof genes in major solanaceous crop under different stress conditions.

	Genes	Functions	Plants	Ref.
**DOF**	*StDof1*	Guard cell specific expression	Potato	[54]
	*NtBBF1*	It controls the vascular development	Tobacco	[55]
	*TDDF1*	It regulates flowering time accelerator, circadian regulation. Act as regulator of different stresses	Tomato	[56]
	*CDF3*	Increases salt tolerance, photosynthetic rate and improves yield (biomass production)	Tomato	[57]
	*StCDF1*	It regulates the tuber formation	Potato	[58]
	*SlCDF1*, *3*	Improves drought, salt and low temperature tolerance. It controls flowering time	Tomato	[58]
	*CaDof17*	Governs the biotic stress tolerance	Pepper	[51]
	*CaDof10*, *11*	Take part in defence against *Phytophthora capsici*, *Pepper mottle virus* and *TMV*	Pepper	[51]

**Table 2 plants-09-00056-t002:** WRKY genes in major solanaceous crop under stress condition.

	Genes	Function in Plants	Plants	Ref.
WRKY	*StWRKY58*	It enhances salt and drought tolerance	Potato	[63]
	*StWRKY1*	It improves tolerance to *Phytophthora infants*	Potato	[64]
	*StWRKY22*	Take part in tolerance to drought, heat and salt stress treatment	Potato	[63]
	*StWRKY8*	Conferring in severe late blight of potato	Potato	[65]
	*SlWRKY58*, *72*	Play role in drought tolerance under water deficiency	Tomato	[62]
	*SlWRKY24*, *37*	Regulates fruit repining and maturity	Tomato	[62]
	*SlWRKY45*	Nematode responsive genes and it has shown resistance to it.	Tomato	[66]
	*SlWRKY39*	Resistance to drought, salt and Pseudomonas syringae pv. tomato DC30000	Tomato	[67]
	*SlWRKY3*	Involved in salt tolerance	Tomato	[68]
	*CaWRKY27*	Positively regulates *Ralstonia solanacearum* infection	Pepper	[69]
	*CaWRKY30*	Pathogen stress response (biotic stress tolerance)	Pepper	[51]
	*CaWRKY58*	Responsible for *Botrytis cinerea* tolerance	Pepper	[51]
	*NtWRKY50*	Take part in resistance to *Ralstonia solanacearum* It changes the level of SA and JA level	Tobacco	[70]
	*NtWRKY6*	Involved in salt and drought stress tolerance	Tobacco	[71]
	*NtWRKY3*, *69*, *70*	Responsible for abiotic stress such as drought and cold	Tobacco	[72]
	*TGA2.2*	Play vital role in Plant defence response and development	Tobacco	[73]

**Table 3 plants-09-00056-t003:** MYB genes in major solanaceous crop with their function.

	Genes	Function in Plants	Plants	Ref.
MYB	*SlMYB14*, *28*, *65*, *66*, *77*, *116*, *22*	Regulate and responsible for SA and JA.	Tomato	[88]
	*SmMYB1* and *6*	It regulates anthocyanin biosynthesis	Eggplant	[84]
	*StMYBA1*	Control the anthocyanin biosynthesis in tobacco	Tobacco	[87]
	*StMYB113*	Regulate the phenylpropanoid metabolism	Arabidopsis	[86]

**Table 4 plants-09-00056-t004:** HSTs genes in major solanaceous crop and their function.

	Genes	Function in Plants	Plants	Ref.
**HsF**	*StHsf4*, *7*, *12*	Responsible for heat, drought and cold stress tolerance.	Potato	[117]
	*SlTCP12*, *15*, *18*	It enhances and regulates fruit ripening.	Potato	[117]
	*StHsf4*, *7*, *9*, *14*	Take part in cold and drought tolerance	Potato	[117]
	*StHsf5*	Play role in heat stress tolerance	Potato	[117]
	*HsfA1a*	Regulates thermotolerance during growth under heat stress	Potato	[118]
	*CaHsfA2*	Govern the thermotolerance and regulates the plant’s ability to resist other environment stresses such as highlight hypoxia, high salt, and osmotic stress	Pepper	[116,119]
	*SlyHSF5*, *7*, *13*, *18*, *23*	It enhances heat stress tolerance	Tomato	[120]
	*SlyHSF-02*	Responsible for triggering the heat response	Tomato	[120]
	*SlHSFA1*	Regulates the thermotolerance in transgenic tomato	Tomato	[121]
	*SlHSFA3*	It Increases thermotolerance and salt hypersensitivity during seed germination in transgenic Arabidopsis	Tomato	[122,123]

**Table 5 plants-09-00056-t005:** NAC genes in major solanaceous crop and their function.

	Genes	Function in Plants	Plant	Ref.
NAC	*SlNAC1*	Take part in salt stress tolerance	Tomato	[136]
		Enhances defense against Pseudomonas infection Chilling and Heat stress tolerance	Tomato	[137]
	*SlNAC4*	Govern salt and drought tolerance, Responsible for fruit ripening and carotenoid accumulation	Tomato	[138,139]
	*SlNAC3*	Control young embryo and endosperm development	Tomato	[140]
	*SlNAP2*	Regulates ABA mediated leaf senescence and help in augmenting fruit yield	Tomato	[141]
	*SlNAC5–10*	It improves salt tolerance (NaCl) treatment	Tomato	[142]
	*JUNGBRUNNEN1*	Responsible for drought stress tolerance	Tomato	[143]
	*SlNAC35*	Govern the root growth and development and Resistance to bacterial pathogen	Tomato	[144]
	*StNAC17*, *30*, *86*, *97*, *85*, *71*	Improves salt and heat stress tolerance	Potato	[145]
	*StNAC2*, *25*, *87*, *91*	Responsible for salt and drought tolerance	Potato	[145]
	*StNAC24*, *59*, *67*, *72*, *108*, *101*	It enhances salt stress tolerance	Potato	[145]
	*StNAC262*	It increases the root size (more lateral roots)	Potato	[146]
	*StNAC4*, *5*, *18*, *48*, *81*	Resistance to *Phytophthora infestans* infection.	Potato	[147]
	*NtNAC1*	Increases the number of lateral roots and nicotine contents.	Tobacco	[148]
	*NtNAC2*	Take part in salt stress tolerance	Tobacco	[148]
	*CaNAC1*	Play crucial role in drought stress and BAX tolerance in *CaNAC1* transgenic tobacco.	Pepper	[150]
	*CaNAC2*	Responsible for cold stress tolerance, root growth and seed maturation.	Pepper	[151]
	*SmNAC*	Increase the susceptibility of plant to bacterial wilt.	Eggplant	[152]
	*SlSRN1*	Response against Biotic Stress (*Botrytis cinerea*)	Tomato	[153]
	*SlNAM2*	Take part in floral whorl and boundary morphogenesis	Pepper	[154]

**Table 6 plants-09-00056-t006:** The studied bZIP genes in potato and tomato under stress condition.

	Genes	Function in Plants	Plants	Ref.
bZIP	*SlbZIP10*, *32*, *33*	It improves plant tolerance to water deficiency(drought)	Tomato	[27]
	*SlbZIP06*, *32*, *46*, *12*, *6*	It regulates phytohormones such as SA, JA and ACC in plants	Tomato	[27]
	*StABF1*	Abiotic stress tolerance (ABA treatment)	Potato	[165]

**Table 7 plants-09-00056-t007:** ERF genes in major solanaceous food crop and their functions.

	Genes	Function in Plants	Plants	Ref.
ERF	*SlERF1*	Responsible for abiotic stress (salt) tolerance and pathogen such as *Botrytis cinerea*, *Xanthomonas campestris and Plectosphaerella cucumerina*. It enhances resistance of tomato fruit to *Rhizopus nigricans*	Tomato	[175,176]
	*SlERF2*	Associated with MeJA-mediated defense and enhance tomato fruit resistance against *Botrytis cinerea*.	Tomato	[177]
	*SlERF3*	Responsible for salt stress and *Ralstonia solanacearum* (biotic stress)		[178]
	*SlERF4*	It plays important role in reduction of ethylene production	Tomato	[177]
	*SlERF84*	Take part in drought and salt stress tolerance. Negatively regulates immunity against *P. syringae pv*.DC3000	Tomato	[179]
	*StERF3*	Responses to abiotic stress such as SA, ABA, and NaCl. Negatively regulate resistance to *Phytophthora infestans*,	Potato	[169]
	*NtERF5*	It enhances the plant resistance to Tobacco Mosaic Virus	Tobacco	[180]
	*StERF37*	Tolerance to abiotic stress such as NaCl, ABA, and heat treatments.	Potato	[181]
	*NtERF114*, *202*, *218*, *228*	It is involved in response to low temperature, drought, and abscisic acid	Tobacco	[72]
	*StERF71*, *47*, *67*, *70*	It is responsible for biotic stress such as *Phytophthora infestans* infected leaves.	Tobacco	[181]
	*TERF1*	Regulate ROS (H_2_O_2_) in tobacco during seedling development.	Tobacco	[182]
	*StERF147*, *169*, *120*, *110*	Responsible for heat stress (temperature above 35 °C)	Tobacco	[181]

**Table 8 plants-09-00056-t008:** ARF genes in major solanaceous food crops and their functions.

	Genes	Function in Plants	Plants	Ref.
ARF	*SlARF3*	It plays multiple roles in the formation of epidermal cells and trichomes	Tomato	[187,188]
	*SlARF7*	It acts as a negative regulator of fruit set until pollination and fertilization	Tomato	[186]
	*SlARF10*	Involved in chlorophyll and sugar accumulation	Tomato	[189]
	*CaARF18*	Take part in flower development and fruit set	Pepper	[190]
	*CaARF10*	Regulates auxin-induced leaf expansion	Pepper	[190]
	*SlARF9*	It regulates the cell division during early tomato fruit development	Tomato	[191]

**Table 9 plants-09-00056-t009:** Number of major transcription factor in solanaceous food crops.

Solanaceous Crops	Transcription Factors						
	WRKY	HSF	MYB	NAC	DOF	ERF	ARF
Potato	79	27	-	110	35	210	20
Tomato	83	26	127	101	34	146	17
Hot pepper	71	25	91	106	33	175	22
Eggplant	50	-	73	-	29		

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
