# Peer review of "The Role of Major Transcription Factors in Solanaceous Food Crops under Different Stress Conditions: Current and Future Perspectives"

_plants, 2020, doi:10.3390/plants9010056_

Round 1
Reviewer 1 Report
The manuscript “The role of major transcription factors in solanaceous food crops under different stress conditions: Current and Future perspectives” summarized data on main groups of transcription factors (TFs) important for plant adaptation to different types of biotic and abiotic stressors. Authors focused on specificity of TFs in Solanaceae because of representation of an important food crops among this family. The review of up-to-date concepts on different mechanisms of crop plants adaptation at transcriptional level to unfavorable environment might highlight direction for further fundamental and practice investigations.
However the manuscript raises several important questions and few comments to be considered and answered:
1. Authors choose DOF, WRKY, MYB, NAC, bZIP and HSF among transcription factors, known to be involved in biotic and abiotic responses. Two additional groups (ARFs and CBFs) are noticed in the Table. Nothing is mentioned about such important groups of stress-related TFs as ERFs and MYCs. Thus it is important to complete the data or at least explain why such a choice was done for Solanaceae.
2. Special attention is paid to new modern genetic approaches, like CRISPR/Cas9 system, to elucidate TF function, but this section is not really in the frame of manuscript and looks a bit excessive.
3. Almost the same could be addressed to the last section – “Osmoprotectant, chlorophyll and relative water content”. Authors did not build a link between earlier discussed TFs, these substances and Solanaceae.
4. It is hard to agree with authors that proline, glycine betaine and others are in the group of osmosensors (Figure 1, block 2). In the same Figure in the block “Gene activation” only in few cases genes are mentioned but not formatted in italics.
Some additional work is needed to correct Tables. For example in Table 1 graph names have to be added as well as genes designation throughout the tables has to be uniform. In Tables 1-6 a left column duplicated table’s titles. Authors are supposed to check citation in text and references list. Finally I would suggest certain English correction. Gerund forms are too often used in the text and number of failed expressions as well. For example: “highly involvement”, “in very specific way or manner”, “structurally speaking “, etc. Also, it is very interesting how authors distinguish between “metabolic” and “biochemical” processes.
Taken together it has to be concluded that the aim of the manuscript is useful and matches interests of the Journal but it could not be published at present form, it needs serious improvement and English correction.
Author Response
Dear reviewer 1, thank you very much for your comments and suggestions. I have put the following answer for each comments.
1. The selected transcription factors are those studied in the selected solanacous crop. I have checked the suggested transcription factor in solanaceous crop and it is not widely studied under different stresses. Therefore, I have removed the names from the figure to avoid ambiguity.
2. In fact, CRISPR Cas9 is the most recent technology used in studying the function of genes by editing(removing or adding). I have tried to mention it to show that this technology is available for those genes I have listed in the paper. If a reviewer think it is excessive I can remove it but it is very important technology in studying the gene function under abiotic and biotic stresses.
3. It is very interesting suggestion, but i have tried to add it. In fact, I wanted show that especially during gene expression and silencing studies, chlorophyll content, relative water content and proline are used as indicator of how much plant is stressed and ensuring of gene silencing by looking a phenotype. Measuring the concentration of proline and chlorophyll as well as relative water content are very important in understanding the physiological status of plant under stress by comparing with the control plant.
4. Yes, I have made mistake on it, but I have recorrected it now!
I have improved the language as much as possible. If there is mistake hope reviewer can mention it.

Reviewer 2 Report
Dear all,
the submitted manuscript is a review of the principal transcription factors involved in abiotic resistance in Solanaceous plants.
I found the paper well written, and in each section the TFs involved are well listed and highlighted with the current knowledge of their function as the new emerging technique of genome editing, with the future perspective of using these technologies in Solanaceous crops in arid or semi-arid climate with improved performances. I also found the references updated to the current knowledge.
Minor point: be careful about the use of ";" instead of ",".
Author Response
Dear reviweer 2
Thank you for your positive suggestion and comment.
I have changed the punctuation mark mentioned the way it was suggested.

Reviewer 3 Report
The topic of this review is about the role of the major transcription factor families in solanaceous food crops under different stress conditions. This topic has not been recently reviewed and therefore has great potential. The authors state that “Solanaceous crops are highly vulnerable to drought and salt due to their body mass (fruits and tubers), succulence and high water requirement particularly during the reproductive stages i.e. flowering, fruiting and seed development”. Therefore, I had expected that the authors elaborate more how these processes might differ in Solanaceae from other plant species. Unfortunately, this topic is rather neglected (besides in the description of the different transcription factors) and the general part could be applied to any plant species. The part on “reverse genetics” nicely gives an introduction to methods that have now been successfully applied to generate loss-of function or gain of function lines in Solanaceae.
The only figure of this paper shows the “Physiological and molecular mechanism of the plant response to the stresses”. The description in the text does not fit very well with the figure. L. 83 “plant responses to biotic and abiotic stresses” - I do not see any indication of abiotic and biotic stresses in the figure. In the figure primary stresses and secondary stresses are indicated, this is not explained in the text. The gray box deals with sensing and transduction of the signal. Here osmosensors are mentioned but they appear also under gene activation. Please explain. Are Phospholipid cleaving enzymes sensors or transducers? Not all stress responses are based on changes in gene activation, this could be clarified in the figure. Blue box: “Transcription control of stresses” should be stress responses?
In the chapter “Osmoprotectant, chlorophyll and relative water content” I am missing the link to the transcription factors.
It is difficult to summarize from all the data on transcription factors those that are solidly linked to stress responses, here the authors broke the knowledge down to the different protein families, which is probably the est way. The tables on the different transcription factor groups are very helpful - would it also be possible to generate a figure to show which transcription factors are mainly involved in which stresses and how they affect the tolerance? This would greatly improve the understanding of the complexity of the regulation.
In 2019 many studies to stress-related proteins in Solanaceae, genome-wide studies of transcription factor families have been published, perhaps the authors could update some literature as very little data is incorporated from 2019.These studies might expand also Table 7.To my knowledge there is data on the NAC transcription factor JUNGBRUNNEN1 (JUB1) in stress responses which is not discussed.
Sometimes it is not clear why an citation was used. For example, L. 73, for a general statement on transcription factors a paper is cited that deals with “Patents on plant transcription factors” (16).
Line 101 “To some degree, these complex cellular responses can be classified into three different phases during the abiotic stress response: alarm, resistance and exhaustion [19]. The cited paper is in a book from 2012. In other publications I found the following citation: “According to Levitt (1980) and further researchers (see reviewed in Kosová et al., 2011), the following phases of plant stress response can be distinguished: an alarm phase, an acclimation phase, a resistance phase, an exhaustion phase, and a recovery phase or death.”
Language (examples):
130: CRISPR/Cas9 has capable of directly - is capable
Tables: please check for not finished entries (SmNAC Increased susceptibility of plant to...) and remove “it” (it helps...)
correlation problems:
23 ..., transcription factors are an answer in improving crop yield ......
TF cannot be the answer but studies on transcription factors can elucidate.....
48 i.e it can be used as a source of drugs and energy - to what is “it” referring to?
51 Not only abiotic factors; biotic stresses are also severely impact the yield
not only ...but also biotic stresses severely impact
138 understand the functional relationship between gene SlCBF1 and chilling stress tolerance -SlCBF1 protein
208 some genes were not expressed (SmeDof25 and SmeDof4), - if these are genes, then they should be italic
217 The studied Dof genes in major solanaceous crop - were the genes studied o the proteins?
466 osmoprotectants are tremendously important for identifying potential false negative results - are you suggesting that plants actively decide between negative and positive results?
515 Genome-wide studies of transcription factors such as WRKY, MYB, DOF HSF, bZIP and NAC
516 in solanaceous plants have shown great importance in boosting up the yield and different stress
517 tolerance under different environmental stress conditions.
Genome wide studies cannot be responsible for stress tolerance
Singular- plural:
58 Transcriptions factors (TFs), which is also referred to as trans-acting factors, are a proteins163 plant-specific multigene Dof (DNA-binding one finger) transcription factor is...DOF transcription factor family
240 Most importantly, signaling molecules ..... plays
469 an osmoprotectant are a
Author Response
Dear Reviewer 3, thank you for your positive comments and suggestions
we have put the following answers accordingly,
Most of Horticultural crops need more water during flowering, fruiting and seed development because of the fact that the biomass of plants increases during this time ( fruits for tomato, eggplant, hot pepper and tuber for potato). In fact, biomass of the solanaceous crops are by far more than that of most cereals, pulse crops and etc. Therefore, solanacous crops demand more water during their growth and development. As we have studied breeding of horticultural crops, we only selected major horticultural crops which are more demanded in daily dishes worldwide. we have added primary and secondary in the text and underlined biotic stresses under primary stress in the figure to make it clear. Under gene activation in the figure, osmosensors were not mentioned. It is osmoprotectants(Proline, Glycine betaine and others. Osmoprotectant concentration usually shows how much the plant is stressed. Not all stresses are based on transcription control,it right but but is possible to check on the gene expression after the plant get exposed to different level of stresses( upregulation and downregulation). On the link of osmprotectants, chlorophylls and relative water content to transcription factors; we want to show that when the plant get exposed to different stresses the accumulation proline and glycine betaine increases. For instance, during the study of gene expression and silencing using different gene silencing methods like VIGS especially when Phytoene desaturase (PDS) used as a control, it is very important to check the chlorophyll and relative water content to understand how much that genes or transcription factors help the plant under stress conditions. As it was suggested we have added JUNGBRUNNEN1 (JUB1) gene. On the abiotic stress response; alarm, resistance and exhaustion, different paper shows resistance and acclimatization together and exhaustion as recovery or dying. Therefore, i have summarized it as three ( alarm, resistance and exhaustion). If the reviewer think it should be changed, we will change it. Language: we have improved it.
Round 2
Reviewer 1 Report
The revised version of the manuscript “The role of major transcription factors in solanaceous food crops under different stress conditions: Current and Future perspectives” still requires additional discussion and correction. The most controversial subject still concerns:
Two groups of TFs, which are ERF and ARF, are still out of the manuscript.Authors commented that information on these groups of TFs is limited and couldn’t be included in the manuscript. But even quick search revealed a list of data that might be suitable for the manuscript.
Li et al., A tomato ERF transcription factor, SlERF84, confers enhanced tolerance to drought and salt stress but negatively regulates immunity against Pseudomonas syringae pv. tomato DC3000 // Plant Physiol Biochem. 2018 Nov;132:683-695. doi: 10.1016/j.plaphy.2018.08.022
Ashrafi-Dehkordi E et al., Meta-analysis of transcriptomic responses to biotic and abiotic stress in tomato // PeerJ. 2018 Jul 17;6:e4631. doi: 10.7717/peerj.4631
Klay I et al., Ethylene Response Factors (ERF) are differentially regulated by different abiotic stress types in tomato plants // Plant Sci. 2018 Sep;274:137-145. doi: 10.1016/j.plantsci.2018.05.023
Zhang H et al., Ethylene Response Factor TERF1, Regulated by ETHYLENE-INSENSITIVE3-like Factors, Functions in Reactive Oxygen Species (ROS) Scavenging in Tobacco (Nicotiana tabacum L.). // Sci Rep. 2016 Jul 20;6:29948. doi: 10.1038/srep29948
Data on ARF TFs in Solonacea is still rather limited but anyhow it is important to be discussed in the manuscript.
Yu et al., Identification and Expression Profiling of the Auxin Response Factors in Capsicum annuum L. under Abiotic Stress and Hormone Treatments // Int J Mol Sci. 2017 Dec 15;18(12). pii: E2719. doi: 10.3390/ijms18122719
Description of CRISPR/Cas9 technology in the manuscript is still under the question. What does this information give to the manuscript??? Only an example that interfering in the structure of gene, encoding transcription factor, in some cases accelerates plant stress tolerance. In the frame of this manuscript one could expect evaluation of what actually was modified, which domain of TF was “damaged” and finally how it changed in regulatory function? What genes finally were activated or inhibited. What actually was modified in signaling? Otherwise quite a number of reviews and manuscripts were published on this newly developed approach, and even concerning its application in Solanaceae.Wang et al. Horticulture Research (2019) 6:77 https://doi.org/10.1038/s41438-019-0159-x
Wolter et al. BMC Plant Biology (2019) 19:176 https://doi.org/10.1186/s12870-019-1775-1
Wang et al., Scientific Reports (2019) 9:1696 https://doi.org/10.1038/s41598-018-38170-6
Chen et al., Annual Review of Plant Biology (2019) 70:667-697 https://doi.org/10.1146/annurev-arplant-050718-100049
It has to be explained more precisely why the part “Osmoprotectant, chlorophyll and relative water content” seems to be redundant. There is no doubt that proline, glycine betaine, antioxidant enzymes, etc. accumulation is a well-known phenomenon for drought, chilling and other type of osmotic stresses. But the title of the manuscript is “The role of major transcription factors…..”. In this case it has to be revealed a direct link between TFs discussed in the manuscript and these reactions, elevating plant tolerance. Authors correct in the opinion that “TF genes need to be further studied to identify suitable candidate to improve different stress tolerance”. Until this link would not be discovered, it is not really needed to be included information about osmoprotectants in this manuscript. Of course arguments might be that it will shorter the field of the manuscript, but it will be more focused on the maim subject – TFs!
Coming back to English correction, I am very sorry, but it has to be continued. To mark all mistakes would need quite a time, thus I will mention quite a few of them.
- incorrect expressions
409 More than two decades ago today
460 This is very interesting because of the fact that studying the genome-wide gene expression patterns could help the plants adapt to different kinds of environmental stresses by developing the transgenic plant or using different gene silencing technologies.
477 plants tissue parts
514 by pestle and mortal
etc.
Please, pay attention also to references.
- style mistakes
lines 20, 25, 27 repetition of “highlighting”, it would be useful to find synonyms
And this is the only one example.
I can recommend to refer a native speaker to improve the manuscript.
Finally, I am sorry but the manuscript still needs serious correction.
Author Response
Dear reviewer 1,
We have added both ERF and ARF transcription factors to the manuscript, but there was no many information regarding to the stress conditions but we have put what is available. Regarding the CRISPR/Cas9, we have tried to add some information which can help the reader to understand genome editing technology. The reason why we added this genome editing technology(CRISPR/Cas9) is that the reader/researcher can use the all mentioned genes that we have summarized from different papers to develop more tolerant and resistant varieties as climate change is ongoing problem. Osmoprotectants;proline and glycine betaine were removed from the manuscript but we have mentioned how RWC and chlorophyll content used in gene silencing and transgenic line development. Regarding abiotic and biotic stresses, we have mentioned in the manuscript precisely, because each table shows more abiotic and biotic stresses. Language: it was improved.
Round 3
Reviewer 1 Report
I am very glad to admit that authors made a big progress with manuscript preparation. But still I have some minor comments and hope that it will be accepted and corrected as well.
One terminological aspect related to “stress” and “stressor” definition meaning. I believe that stress is an organism's response to a stressor, i.e. stress factor. I kindly ask authors to look through the manuscript and make corrections all over text, including Fig. 1.
Coming to the Fig. 1, some changes are required:
“water scarcity” to “water deficiency” in higher left box at the second point “heat” is mentioned, what is a difference of this stressor from the next point - “extreme temperature” why soil salinity is combined with Biotic stress?
Another aspect is what TFs are actually doing in the cell. It is commonly known that TFs regulates expression of different genes, and thus involved the development of organism response to stressor application. Therefore TFs could not help, aid, support and etc. in synthesis, tolerance and etc. They do regulate, control, govern gene expression, synthesis, processes etc. This expressions need to be changed in text and especially in tables. Newly presented tables for ERFs and ARFs might be taken as an example. But these tables have to be numbered and cited in the text of manuscript.
One little more comment: Am I correct that Key Laboratory of Agricultural Water Resources, Hebie Laboratory of Agricultural Water Saving, Center for Agricultural Resources Research, Institute of Genetics and Developmental Biology, Shijiazhuang, Hebie, 050021 are located in China? Please, correct affiliation.
And last, of course that is the right of the authors to complete the manuscript with different data but I still believe that data on CRISPR/Cas9 and measuring of chlorophyll and relative water content are not really required.
Author Response
Dear reviewer 1
We have corrected the way it was suggested. But regarding the CRISPR/Cas9, we have mentioned in conclusion the importance of this technology in fighting against climate change and developing improved varieties in short scale of time. We thought reverse genetics is the most effective technology in improving yield and developing stress tolerant plant. Measuring physiological indices such as chlorophyll and relative water content are very much important in studying reverse genetics such as gene silencing as a supportive data.
Thank you very much for your positive comments and suggestions all along.